# Bioconversion of Deoxynivalenol-Contaminated Feed by Yellow Mealworm (*Tenebrio molitor*) Larvae in the Production of Biomass Intended for Feed Purposes

**DOI:** 10.3390/toxins17080420

**Published:** 2025-08-19

**Authors:** Marcin Wróbel, Michał Dąbrowski, Michał Łuczyński, Tadeusz Bakuła, Natalia Gruchała, Łukasz Zielonka

**Affiliations:** 1Department of Research and Development, Chemprof, Gutkowo 54B, 11-041 Olsztyn, Poland; marcin.wrobel@chemprof.pl (M.W.); michal.luczynski@chemprof.pl (M.Ł.); 2Department of Veterinary Prevention and Feed Hygiene, Faculty of Veterinary Medicine, University of Warmia and Mazury in Olsztyn, Oczapowskiego 13/29, 10-718 Olsztyn, Poland; michal.dabrowski@uwm.edu.pl (M.D.); bakta@uwm.edu.pl (T.B.); natalia.gruchala@student.uwm.edu.pl (N.G.)

**Keywords:** yellow mealworm, biodegradation, deoxynivalenol (DON), detoxification, PAP (Processed Animal Protein)

## Abstract

Deoxynivalenol (DON) is a prevalent mycotoxin in feed, raising concerns about its impact on animal health and feed safety. Insects such as yellow mealworm (*Tenebrio molitor*) may play a role in the biodegradation of DON-contaminated feed. This study presents the results of a two-week rearing experiment, where 19-week-old yellow mealworm larvae were fed diets with varying concentrations of DON. The larvae were divided into three groups (C, A, and B) that differed in the amount of mycotoxin added to the feed. Larval survival, body mass, and DON accumulation in larvae and their frass were evaluated. A statistical analysis revealed no significant differences in larval survival or body mass gain between the groups. The results point to the low accumulation of DON in larvae, reaching 13.13 ± 2.06 µg/kg (A) and 32.18 ± 4.20 µg/kg (B) after two weeks of feeding. Moreover, at the end of the experiment, DON was detected in larval frass at high concentrations of 507.65 ± 15.31 µg/kg (A) and 862.61 ± 18.53 µg/kg (B), suggesting that larvae are capable of effectively excreting this mycotoxin. The analyzed mycotoxin had no significant effect on larval survival or growth. Deoxynivalenol did not accumulate in the larval biomass and was excreted with frass. These findings enhance our understanding of the interactions between DON and yellow mealworm larvae and have potential implications for using insects in feed production and mycotoxin neutralization within ecosystems. *Tenebrio molitor* larvae tolerate DON-contaminated feed and effectively excrete the toxin, making them potential candidates for feed detoxification systems.

## 1. Introduction

Mycotoxins are secondary metabolites produced by filamentous fungi that exert toxic effects on animals. These fungi can colonize crops during growth or agricultural products during storage. Mycotoxins pose a serious threat to the health and welfare of both humans and animals and raise significant food safety concerns [1]. This large group of chemical compounds can induce a wide range of adverse health effects in both animals and humans [2].

Deoxynivalenol (DON), a type-B trichothecene commonly known as “vomitoxin”, is one of the most prevalent mycotoxins [3]. This toxin is produced mainly by pathogenic fungi such as *Fusarium graminearum* and *Fusarium culmorum* that colonize cereals, including wheat, maize, rye, barley, oats, and rice, in temperate climates. Deoxynivalenol is often accompanied by its acetylated derivatives, including 3-acetyl-DON (3-Ac-DON) and 15-acetyl-DON (15-Ac-DON) [4,5].

Deoxynivalenol is considered a thermostable compound that can tolerate temperatures of up to 350 °C and is difficult to eliminate through thermal processing [6,7]. In the European Union, the guidance values for DON have been set at 0.9–12 mg/kg in animal feed [8] and at 0.15–1.75 mg/kg in food intended for human consumption [9]. The International Agency for Research on Cancer (IARC) has classified DON as a Group 3 substance, which implies that it is not classifiable as to its carcinogenicity to humans [10]. However, DON can cause acute gastrointestinal symptoms, such as vomiting, in both animals and humans [4].

Animal studies have demonstrated that chronic DON exposure disrupts protein synthesis, impairs body mass gain, reduces feed efficiency, and negatively impacts the immune system [1,11]. Animal species differ in their sensitivity to DON. Monogastric animals, particularly pigs, are most susceptible, whereas poultry and ruminants are more tolerant [3,12].

Deoxynivalenol is most prevalent in temperate regions, including Europe (especially Northern and Central Europe), North America, East Asia, and China [13,14,15,16,17,18,19,20]. According to the literature and global research, animal feeds are widely contaminated with mycotoxins and regional trends are often observed, subject to climate and weather [20]. For instance, rainfall is a key factor influencing DON contamination levels in maize [13]. Favorable conditions for *Fusarium* growth include warm temperatures (21–25 °C) and high humidity. Grains stored under moist conditions are particularly susceptible to damage [12]. *Fusarium* infections can be prevented through proper crop management and rapid post-harvest drying, which significantly reduce DON contamination in grain [6]. The highest DON levels are typically found in unprocessed wheat, maize, oats, and their by-products [4].

Mycotoxin contamination of animal feed poses a significant challenge in the livestock industry and a serious threat to animal health. Although various decontamination methods exist, they are not always sufficiently effective and may incur additional costs [12,21].

In Commission Regulation (EU) No 2017/893 of 24 May 2017, seven insect species, including yellow mealworm (*Tenebrio molitor*), have been recognized as farmed animals. This regulation permits the use of insect-derived Processed Animal Protein (PAP) in feed for aquaculture, carnivorous fur animals, and livestock [22]. Since August 2021, the use of insect-derived PAP has also been allowed in pig and poultry nutrition [23]. Furthermore, Regulation (EU) 2022/169 of 8 February 2022 authorized the marketing of frozen, dried, and powdered yellow mealworm larvae (*T. molitor*) as a novel food for humans [24].

The nutritional value of insects varies depending on the developmental stage, production conditions, and feed composition. According to the literature, *T. molitor* larvae are a rich source of protein with a favorable amino acid profile. Dried larvae may contain over 50% protein and approximately 30% fat, making them a good source of energy. Larval fat is rich in unsaturated fatty acids, particularly oleic and linoleic acids, as well as essential amino acids such as leucine, lysine, tyrosine, and valine [25,26,27].

Industrial insect production is an economically viable, energy-efficient, and environmentally friendly method of protein production. It causes less environmental pollution than traditional livestock farming [28]. Yellow mealworms are capable of effectively bioconverting low-quality feed and plant-based waste into high-value, protein-rich biomass [25].

In a review article, Vandeveyer [29] concluded that the use of *T. molitor* larvae as a feed ingredient entails substantial biological safety risks, particularly those associated with pathogenic bacteria and mycotoxins. Spore-forming bacteria such as *Bacillus cereus*, *Clostridium* spp., and *Staphylococcus aureus* represent the most frequently reported microbial hazards in mealworms. These microorganisms may persist in the larval gut or as spores, making them difficult to eliminate through conventional processing methods.

Recent advances in multi-omics approaches have shown that dietary composition, gut microbiota, and host metabolism are closely interconnected, influencing nutrient utilization and xenobiotic detoxification processes [30]. Such integrative perspectives may also be relevant to understanding how *T. molitor* larvae metabolize or tolerate DON-contaminated feed. Considering the chemical stability of mycotoxins and the resilience of bacterial spores, preventive measures such as thermal treatments, species–specific risk assessments, and continuous microbiological monitoring should be implemented throughout the production process to mitigate risks. 

Therefore, this study was undertaken to evaluate the suitability of low-quality feed contaminated with DON for rearing *T. molitor* larvae intended for the production of animal feed. The aim of the experiment was to determine whether DON-contaminated diets adversely affect rearing parameters and whether toxin accumulation levels limit the applicability of the resulting biomass for feed production.

## 2. Results

### 2.1. Effect of Deoxynivalenol on Insect Development

At the end of the first week of the experiment, the average number of dead larvae was 101.00 ± 15.13 in the control group (C), 88.33 ± 6.51 in group A, and 100.00 ± 30.61 in group B. However, no significant differences were observed between the groups (*p* > 0.05). After two weeks of feeding, no significant differences in mortality were observed between groups C, A, and B. The total number of dead larvae collected at the end of the experiment was 190.67 ± 32.52 (C), 172.67 ± 8.14 (A), and 158.67 ± 10.41 (B).

An analysis of larval metamorphosis into pupae and adult insects conducted at the end of the first week revealed 24.33 ± 7.57 (C), 12.00 ± 5.00 (A), and 15.67 ± 7.64 (B) transformed individuals. No adult insects were observed at this stage. No significant differences were found between the groups (*p* = 0.16). Adult insects began to emerge in the second week of the study. At the end of the experiment, the total number of pupae and adult insects reached 83.67 ± 12.01 (C), 103.33 ± 2.08 (A), and 103.33 ± 8.62 (B). Although ANOVA yielded a borderline *p*-value (*p* = 0.05), Tukey’s HSD test did not reveal significant differences between the groups (Table 1).

The initial average body mass of larvae used in the experiment was 112.47 ± 23.95 mg. After one week of feeding, the larval body mass increased by 41.70% in group C, 36.69% in group A, and 41.54% in group B. The post hoc analysis revealed no significant differences between the groups (*p* = 0.13). After two weeks, the larval biomass increased by 74.78% (C), 79.74% (A), and 74.93% (B). Once again, Tukey’s HSD test did not reveal significant differences between the groups (*p* = 0.05) (Table 2).

### 2.2. Deoxynivalenol Accumulation in the Biomass of Yellow Mealworm Larvae

The mean concentration of DON in larvae, determined by LC-MS/MS (Table 3), reached 1.83 ± 0.41 µg/kg in the control group after one week and 5.18 ± 2.9 µg/kg after two weeks. In group A larvae (administered diets containing 663 µg DON/kg), DON levels were determined at 6.30 ± 1.45 µg/kg after one week and 13.13 ± 2.06 µg/kg after two weeks. In group B larvae (administered diets containing 919 µg DON/kg), DON levels reached 15.57 ± 1.85 µg/kg and 32.18 ± 4.20 µg/kg, respectively.

At the end of the experiment, the transfer coefficient of DON from feed to larval biomass reached 1.98 ± 0.31% in group A and 3.50 ± 0.46% in group B. Significant differences were found between the groups in both weeks of the experiment (*p* < 0.05).

### 2.3. Deoxynivalenol Levels in Larval Frass

Deoxynivalenol levels in larval frass were high (Table 4). At the end of the first week, the DON concentration was determined at 400.05 ± 18.53 µg/kg in group A and 689.15 ± 38.18 µg/kg in group B, accounting for 60.34 ± 2.80% and 74.99 ± 4.15% of the original values in feed, respectively. After two weeks, DON levels further increased to 507.65 ± 15.31 µg/kg in group A (76.57 ± 2.31%) and 862.61 ± 18.53 µg/kg in group B (93.86 ± 2.02%). Significant differences between groups were found in both weeks of the experiment (*p* < 0.05).

## 3. Discussion

### 3.1. Larval Development

The experiment revealed a satisfactory growth performance of larvae reared on the analyzed feed. After two weeks of feeding, the average increase in larval body mass ranged from 74.93% to 79.74%. The results of the larval mortality analysis and the number of pupae and emerged adults indicate that the presence of DON in the diet had no negative effect on body mass gain or larval survival.

Similar findings were reported by other authors who observed that DON concentrations of up to 12,000 µg/kg did not compromise the growth of *T. molitor* larvae in terms of body mass and average daily gain (ADG) [31]. Likewise, Van Broekhoven et al. [32] found no differences in larval development between larvae fed a control diet and those fed naturally contaminated or DON-supplemented diets.

Larvae exposed to fumonisin B1 initially grew at rates comparable to the control group. A decrease in the larval growth rate was noted only after 28 days of feeding, which was indicative of slow, chronic intoxication, especially in response to the highest toxin levels [33]. Other studies have also shown that exposure to aflatoxin B1 (AFB1) for 10 days had no significant impact on larval survival or body mass regardless of the administered dose, which confirms that *T. molitor* has a high tolerance to AFB1 [34].

Similarly, no significant increase in larval mortality was observed over a 50-day feeding period in experiments involving AFB1, ochratoxin A (OTA), and fumonisin B1 (FB1) [35]. These results suggest that short-term exposure to moderate concentrations of DON and other mycotoxins does not impair the development of *T. molitor*.

However, Jankovic-Tomanic et al. [36] reported a decrease in the body mass of larvae administered DON-contaminated feed for two weeks, and this effect was intensified as DON concentrations increased. The greatest decrease was noted in groups exposed to very high DON levels (16–25 mg/g). These findings point to the presence of a threshold concentration beyond which larval detoxification mechanisms are overwhelmed, resulting in impaired growth and survival.

### 3.2. Deoxynivalenol in Larvae

After two weeks of feeding DON-contaminated diets, the maximum DON concentration in larvae reached 32.18 ± 4.20 µg/kg, corresponding to a transfer rate of only 3.5%. This concentration is well below the maximum limits established for both food and feed in current regulations [8,9].

However, DON levels in larvae increased proportionally with both the concentration of DON in feed and the duration of exposure. Therefore, prior to further processing, it is advisable to fast larvae for at least 24 h. This practice facilitates the elimination of contaminated feed from the digestive tract and likely enhances detoxification and the excretion of DON. In other animal models, metabolomic analyses have demonstrated that DON exposure can disrupt amino acid metabolism and sphingolipid signaling pathways, contributing to intestinal toxicity and metabolic stress [37]. Although the physiological architecture of *T. molitor* differs substantially from vertebrates, similar biochemical pathways may be involved in DON detoxification or tolerance, warranting further targeted metabolomic studies in insects. Similar fasting protocols were found to be effective in previous studies involving zearalenone, ochratoxin A, and T-2 toxin, resulting in rapid reductions in mycotoxin residues in larval tissues [38].

The content of AFB1, OTA, and FB1 in larvae was reduced by 57–100% after a 24 h detoxification period, achieving levels considered safe for consumption [35]. Overall, the available literature confirms that *T. molitor* larvae demonstrate a high tolerance to DON and other mycotoxins.

For example, Bosch et al. [34] showed that yellow mealworm larvae tolerated AFB1 levels of up to 0.415 mg/kg in dry feed. After the administration of AFB1-contaminated feed, larvae retained only trace amounts of the toxin, corresponding to approximately 10% of the EU limit for feed materials (0.02 mg/kg) [39]. In another study where larvae were fed wheat bran contaminated with 50–200 µg/kg of AFB1, conversion rates reached up to 87.85%, while residual AFB1 levels in larvae remained below 2 µg/kg [40].

In another experiment, larvae reared for four weeks on feed naturally contaminated with DON accumulated no more than 131 µg/kg of the mycotoxin in their biomass, despite the fact that the DON concentration in the feed reached up to 12,000 µg/kg. Deoxynivalenol levels were as high as 200 µg/kg even in the control group’s feed [31]. In the work of Van Broekhoven et al. [32], DON was not detected in the larval biomass, but the detection limit of the method used was 30 µg/kg.

These findings suggest that lower-quality feed containing moderate levels of DON can be used to produce safe larval biomass for feed purposes. It has been demonstrated that larvae reared on wheat contaminated with up to 30,730 µg/kg of DON can be used as an effective protein source in poultry diets [41]. Therefore, it appears that the safety thresholds for DON in *T. molitor* feed may be higher than those applied in conventional livestock production.

A similar conclusion was drawn in a study of *Alphitobius diaperinus*, another tenebrionid beetle species that was fed a diet containing a mixture of AFB1, DON, zearalenone, and OTA at concentrations exceeding regulatory limits by up to 25 times. Researchers found no dangerous accumulation of these toxins in the larval biomass [42]. This is particularly relevant, as feed is often contaminated with multiple mycotoxins.

### 3.3. Deoxynivalenol in Frass

The presence of high DON levels in larval frass was one of the key findings of the experiment. After two weeks of feeding contaminated diets, the DON concentration in frass reached up to 862.61 ± 18.53 µg/kg, which implies that up to 93.86 ± 2.02% of the ingested DON was excreted.

Deoxynivalenol levels in frass were directly correlated with dietary toxin concentrations and increased on successive days of the experiment. A similar trend was observed in a study by Carlos Ochoa Sanabria et al. [31], where DON levels in frass peaked at 742 µg/kg in larvae fed the highest DON concentrations. The presence of DON in larval frass was also reported by Van Broekhoven et al. [32]. Their study demonstrated that the percentage of excreted DON was higher in *T. molitor* larvae fed enriched flour (41%) than naturally contaminated flour (14%). Similar results were reported in larvae exposed to other mycotoxins. For example, depending on the initial concentration, 20–35% of dietary AFB1 was directly excreted by the larvae [40].

Abado-Becognee also reported that *T. molitor* larvae excreted between 38% and 42% of fumonisin B1 with frass [33]. These findings suggest that the low absorption of DON in the digestive tract is one of the main tolerance mechanisms in *T. molitor*.

Further research is needed to explore the pathways responsible for mycotoxin metabolism in *T. molitor*. These insects may also possess metabolic routes that degrade DON into less toxic or undetectable compounds that cannot be detected with standard analytical techniques.

## 4. Conclusions

The analysis of the experimental results and the literature data suggests that *T. molitor* larvae exhibit a relatively high resistance to DON in feed. No apparent negative effects on larval development were observed during short-term exposure to the mycotoxin. Additionally, the absence of significant DON accumulation in larval biomass indicates that the obtained material can be further processed into animal feed.

These findings imply that *T. molitor* larvae can be used for the bioconversion, biodegradation, and detoxification of mycotoxin-contaminated feed. Their ability to tolerate feed contaminants and maintain normal development in the presence of toxins suggests that yellow mealworms could be an effective biological detoxification tool in the feed and agricultural industries.

## 5. Materials and Methods

### 5.1. Chemicals

The DON standard for chromatographic analysis and feed preparation was obtained from Sigma-Aldrich (Poznań, Poland). LC-MS-grade acetonitrile, methanol, water, and the reagents used for chromatography, extraction, and mobile phase preparation, including ammonium formate (NH_4_COOH), formic acid (FA), and ammonium fluoride (AF), were purchased from Merck (Poznań, Poland).

### 5.2. Feeding Substrates and Diet Preparation

Two experimental diets with different concentrations of DON were prepared. The basal feed contained 23.64% protein, 5.28% crude fat, 37.68% starch, 3.99% crude fiber, 6.64% ash, 10.61% moisture, and DON at a concentration of 28.71 µg/kg. The composition of basal feed was determined using a Foss InfraXact™ analyzer (FOSS Poland, Warszawa, Poland).

To prepare diets A and B, DON (Sigma Aldrich, Poznań, Poland) was dissolved in methanol and sprayed onto the basal feed. The feed was thoroughly mixed and dried at approximately 50 °C under reduced pressure. Samples from various locations within the batch were collected to assess homogeneity and DON content in the LC-MS/MS assay. The final compositions of the experimental diets are shown in Table 5. The elemental composition of the base feed was determined using a near-infrared (NIR) spectroscopy-based analyzer (Foss InfraXact™). While this method allows for rapid and non-destructive analysis, it does not account for all chemical constituents of the feed, particularly some nitrogen-containing compounds and specific carbohydrate fractions. Therefore, the sum of the quantified components does not reach 100%, and the missing fraction likely includes undetectable minor constituents.

### 5.3. Feeding Experiment

Nineteen-week-old *T. molitor* larvae with an average body mass of 112.47 ± 23.95 mg were supplied by TENEBRIA Sp. z o.o., Lubawa, Poland, and randomly assigned to three experimental groups. Each group had six replicates with approximately 50 g of larvae per replicate. The larvae were housed in polypropylene containers (31 × 17 × 9.5 cm).

Group C: Control group administered basal feed;Group A: Administered feed contaminated with 663 µg DON/kg;Group B: Administered feed contaminated with 913 µg DON/kg.

Larvae were reared for two weeks in a Memmert HPP 749 incubator at 27 ± 2 °C and 70 ± 5% relative humidity. Half of the larvae (three containers per group) were analyzed after the first week (C1, A1, and B1) and the remaining half after the second week (C2, A2, and B2).

Larvae were separated from unconsumed feed and frass using 0.5 mm sieves. Larvae were fasted for 24 h to cleanse the gut. Fasted larvae were weighed and euthanized by freezing at −24 °C.

Throughout the experiment, larvae were fed a mixture composed of 80% dry feed (control or DON-contaminated) and 20% fresh carrot pulp as a water source. Larvae were fed once daily at 5:00 p.m. Dead larvae were counted daily, and the number of pupae and adult insects was recorded at the end of each week.

The amount of the administered feed was increased daily in proportion to larval growth, as shown in Table 6.

### 5.4. Extraction of Deoxynivalenol from Larvae and Frass

Approximately 5 g of frozen larvae were ground for 30 s in a Millmix 20 ball mill (Millmix Agro, Madona, Latvia) at a vibrating frequency of 20 Hz. One gram of the ground sample was transferred to a 15 mL Falcon tube, and 5 mL of 80% acetonitrile with 1% formic acid was added. The samples were sonicated for 30 s in a Sonics Vibra Cell (Sonics, Singapore) at 40% amplitude.

The samples were then shaken in an Eberbach shaker (model EL.680.Q.25) for 30 min at 400 oscillations/min and centrifuged at 10,000 rpm for 10 min at 4 °C (Eppendorf Centrifuge 5804 R, Eppendorf, Hamburg, Germany). The supernatants were filtered through 0.22 µm Teflon syringe filters into LC vials and analyzed by LC-MS/MS. Each sample was prepared in triplicate. The same procedure was applied to frass samples.

### 5.5. Deoxynivalenol Analysis by LC-MS/MS

Deoxynivalenol concentrations in feed, larvae, and frass were measured using liquid chromatography coupled with tandem mass spectrometry (LC-MS/MS) in an Agilent 1260 Infinity II chromatograph connected to an Agilent 6470 Triple Quadrupole detector (Agilent Technologies, Santa Clara, CA, USA). A modified version of the method developed by Paschoal was used in the extraction and chromatographic analysis [43].

Separation was performed at a flow rate of 0.5 mL/min using an Agilent Poroshell 120 SB-C18 column (Agilent, Santa Clara, CA, USA, 2.7 µm, 2.1 × 100 mm). The mobile phase consisted of the following:

Phase A: H_2_O with 0.2% FA, 5 mM ammonium formate, and 0.5 mM ammonium fluoride;

Phase B: Methanol with 0.5% FA and 0.5 mM ammonium fluoride.

The following gradient elution method was used:0–1 min: 10% B;1–2 min: Increase to 50% B;2–10 min: Linear increase to 95% B;10–14 min: Hold at 95% B;14–18 min: Return to 10% B for re-equilibration.

The samples were injected using a sandwich technique (40 µL H_2_O + 2.5 µL sample). The column temperature was maintained at 40 °C. The injection needle was rinsed with 50% methanol for 3 s. The following ion source parameters were used: ESI + AJS, gas temp—250 °C, gas flow—8 L/min, nebulizer—45 psi, sheath gas temperature—350 °C, flow rate—11 L/min, and capillary voltage—3300 V.

Deoxynivalenol was monitored in the multiple reaction monitoring (MRM) mode using *m/z* 297.2 → 249.2 (quantifier) and *m/z* 297.2 → 203.2 (qualifier) transitions. Data were acquired and processed using Agilent MassHunter software (version 10.1). The detection limit (LOD) for DON in larvae and frass was approximately 2 ng/g and the limit of quantification (LOQ) was 6 ng/g.

### 5.6. Statistical Analysis

Data were analyzed using Statistica 13.3 (TIBCO Software Inc., Palo Alto, CA, USA). The normality and homogeneity of variance were assessed using the Shapiro–Wilk test and Levene’s test, respectively. If the assumptions of normality and equal variances were met, larval mass, survival, and adult/pupa counts were analyzed by one-way ANOVA with Tukey’s post hoc test (*p* < 0.05).

The concentration of DON in larvae and frass was determined using the non-parametric Kruskal–Wallis test (*p* < 0.05). All values were expressed as means ± standard deviation (SD).

## Figures and Tables

**Table 1 toxins-17-00420-t001:** Survival of *Tenebrio molitor* larvae during the experiment. Letter (a) indicates significant differences (*p* < 0.05).

Group	Average Number of Dead Larvae		Average Number of Pupae and Adults	
	Week 1	Week 2	Week 1	Week 2
Control (C)	101.00 ± 15.13 ^a^	190.67 ± 32.52 ^a^	24.33 ± 7.57 ^a^	83.67 ± 12.01 ^a^
A	88.33 ± 6.51 ^a^	172.67 ± 8.14 ^a^	12.00 ± 5.00 ^a^	103.33 ± 2.08 ^a^
B	100.00 ± 30.61 ^a^	158.67 ± 10.41 ^a^	15.67 ± 7.64 ^a^	103.33 ± 8.62 ^a^

**Table 2 toxins-17-00420-t002:** Average body mass of *Tenebrio molitor* larvae administered deoxynivalenol (DON)-contaminated diets for one and two weeks. Letter (a) indicates significant differences (*p* < 0.05).

Group	Average Larval Body Mass (mg)		Increase in Larval Body Mass (%)	
	Week 1	Week 2	Week 1	Week 2
Control (C)	159.37 ± 4.40 ^a^	196.58 ± 1.38 ^a^	41.70	74.78
A	153.73 ± 0.83 ^a^	202.15 ± 3.41 ^a^	36.69	79.74
B	159.19 ± 3.30 ^a^	196.74 ± 1.87 ^a^	41.54	74.93

**Table 3 toxins-17-00420-t003:** Deoxynivalenol (DON) levels (determined by LC-MS/MS) in yellow mealworm larvae (*Tenebrio molitor*) administered DON-free or DON-contaminated diets. Different letters (a, b, and c) indicate significant differences (*p* < 0.05).

Group	DON (µg/kg)		Transfer Coefficient from Feed to Larval Biomass (%)	
	Week 1	Week 2	Week 1	Week 2
Control (C)	1.83 ± 0.41 ^c^	5.18 ± 2.9 ^c^	-	-
A	6.30 ± 1.45 ^b^	13.13 ± 2.06 ^b^	0.95 ± 0.22	1.98 ± 0.31
B	15.57 ± 1.85 ^a^	32.18 ± 4.20 ^a^	1.69 ± 0.20	3.50 ± 0.46

**Table 4 toxins-17-00420-t004:** Concentration and percentage of deoxynivalenol (DON) (determined by LC-MS/MS) excreted by *Tenebrio molitor* larvae. Different letters (a, b, and c) indicate significant differences (*p* < 0.05).

Group	DON (µg/kg)		Transfer Coefficient from Larval Biomass to Frass (%)	
	Week 1	Week 2	Week 1	Week 2
Control (C)	10.56 ± 6.74 ^c^	7.40 ± 3.22 ^c^	-	-
A	400.05 ± 18.53 ^b^	507.65 ± 15.31 ^b^	60.34 ± 2.80	76.57 ± 2.31
B	689.15 ± 38.18 ^a^	862.61 ± 18.53 ^a^	74.99 ± 4.15	93.86 ± 2.02

**Table 5 toxins-17-00420-t005:** Proximate composition of the experimental diets.

Ingredient	Content [%]		
	Diet C	Diet A	Diet B
Ash	6.64		
Crude fiber	3.99		
Starch	37.68		
Protein	23.64		
Fat	5.28		
Moisture	10.61		
	Mycotoxin concentration (μg/kg)		
Deoxynivalenol	28.71	663	919

**Table 6 toxins-17-00420-t006:** Daily increase in the dose of dry feed.

Day	Daily Amount of Dry Feed (g)
1	15.63
2	17.58
3	19.78
4	22.25
5	25.02
6	28.16
7	31.68
8	35.64
9	40.09
10	45.10
11	50.74
12	57.08
13	64.22
14	72.24

## Data Availability

Dataset available on request from the authors.

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
