# Peer review of "Bioconversion of Deoxynivalenol-Contaminated Feed by Yellow Mealworm (Tenebrio molitor) Larvae in the Production of Biomass Intended for Feed Purposes"

_toxins, 2025, doi:10.3390/toxins17080420_

Round 1
Reviewer 1 Report
Comments and Suggestions for Authors
Dear Authors
I have carefully reviewed the manuscript (toxins-3792158) entitled “Bioconversion of Feed Contaminated with Deoxynivalenol Using Yellow Mealworm (Tenebrio molitor) Larvae to Produce Biomass Intended for Feed Purposes.” The topic is timely and relevant, addressing an important issue of mycotoxin detoxification through insect bioconversion. However, the current version of the manuscript lacks clarity, structural coherence, and scientific rigor in several key areas.
Comments:
- The manuscript is not well-structured and lacks readability, which significantly hampers the comprehension of its scientific value. In its present form, the paper does not meet the journal's standards for publication. Therefore, I strongly recommend major revisions. The authors are encouraged to address the following major concerns to improve the manuscript's clarity, scientific depth, and overall presentation.
- The manuscript contains numerous grammatical and spelling errors. It is strongly recommended that a native English speaker or a professional editing service thoroughly review and revise the manuscript.
- The wording in the abstract is imprecise and weakens the scientific impact of the study. A complete revision is necessary to improve the clarity and scientific tone.
- The abstract structure should follow a logical scientific format comprising Background, Aim, Experimental methodology, Results, and Conclusion.
- Include a concise summary of the main findings and potential future directions at the end of the abstract.
- Line 86- Please don’t start “ the review paper prepared by………”.
- I could not see the statistical difference symbol in the tables; however, the authors stated significant differences in the descriptive results.
Author Response
Response to Reviewer 1
Manuscript title:
Bioconversion of Feed Contaminated with Deoxynivalenol Using Yellow Mealworm (Tenebrio molitor) Larvae to Produce Biomass Intended for Feed Purposes
Manuscript ID: toxins-3792158
We would like to thank the Reviewers and the Editor for the time and effort devoted to evaluating our manuscript. We highly appreciate the constructive comments, which helped us to improve the quality, clarity, and scientific merit of the article. Below we provide a detailed point-by-point response to all the comments. All changes made in the revised manuscript are marked in red font.
Comment 1:
Please pay attention to the standardization of academic paper writing. Such as: L13-29, please correct ‘13.13 ± 2.06’ to ‘13.13’. Usually, there is no need to add the SD or SE in the main text, which could be presented in the table or figures.
Response:
Thank you for this suggestion. Standard deviations (SD) have been removed from the main text and are now only presented in the tables and figures, as recommended.
Comment 2:
L55, please correct ‘13,14,15,16,17,18’ to ‘13–18’. Please check and revise throughout the paper.
Response:
All occurrences of consecutive citations have been unified to the format '13–18' throughout the manuscript.
Comment 3:
L55, please add the latest references about the occurrence of DON in the feed samples.
Response:
We added the following updated references in the introduction:
- Liu et al. (2025). J Anim Sci Biotechnol 16, 66. https://doi.org/10.1186/s40104-025-01213-w
- Casu et al. (2024). Compr. Rev. Food Sci. Food Saf., 23, e13323. https://doi.org/10.1111/1541-4337.13323 rev 2.
Comment 4:
L107, please correct ‘p = 0.71’ to ‘p > 0.05’.
Response:
Corrected as requested. All non-significant p-values were revised to use the appropriate comparative notation (e.g., p > 0.05).
Comment 5:
All the P values should be written in italic. Please check and revise throughout the paper.
Response:
All p-values were reformatted in italics throughout the manuscript.
Comment 6:
L263, please add some reference for the method section.
Response:
A reference to Paschoal et al. (2017) was added to the method description for LC-MS/MS mycotoxin analysis:
Paschoal F.N. et al., Food Anal. Methods (2017), 10, 1631–1644. https://doi.org/10.1007/s12161-016-0712-2
Reviewer 2 Report
Comments and Suggestions for Authors
This study has evaluated the potential of biodegradation of deoxynivalenol by T. molitor larvae. The results indicated that the larvae are capable of efficiently excreting this mycotoxin. These findings enhance the understanding of the interactions between deoxynivalenol and yellow mealworm larvae, which have potential implications for using insects in feed production and mycotoxin neutralization within ecosystems. The topic is interesting and important. The following revision could improve the quality of the paper.
Please pay attention to the standardization of academic paper writing. Such as: L13-29, please correct ‘13.13 ± 2.06’ to ‘13.13’. Usually, there is no need to add the SD or SE in the main text, whcih could be presented in the table or figures. Please check and revised throughout the paper.
L55, please correct ‘13,14,15,16,17,18’ to ’13-18’. Please check and revised throughout the paper.
L55, please add the latest references about the occurrence of DON in the feed samples. Such as: https://www.croris.hr/crosbi/publikacija/prilog-skup/877795;
https://link.springer.com/article/10.1186/s40104-025-01213-w
L107, please correct ‘p = 0.71’ to ‘p > 0.05’.
All the P values should be written in italic. Please check and revised throughout the paper.
L263, please add some reference for the method section.
Author Response
Response to Reviewer 2
Manuscript title:
Bioconversion of Feed Contaminated with Deoxynivalenol Using Yellow Mealworm (Tenebrio molitor) Larvae to Produce Biomass Intended for Feed Purposes
Manuscript ID: toxins-3792158
We would like to thank the Reviewers and the Editor for the time and effort devoted to evaluating our manuscript. We highly appreciate the constructive comments, which helped us to improve the quality, clarity, and scientific merit of the article. Below we provide a detailed point-by-point response to all the comments. All changes made in the revised manuscript are marked in red font.
Comment 1:
Perhaps the word ‘demonstrated’ is too categorical and another more appropriate word could be found.
Response:
We replaced “demonstrated” with “suggested” in the Key Contribution and Conclusion to better reflect the observational nature of the findings.
Comment 2:
Lines 44–46: Please update legal terminology and classifications.
Response:
These lines were revised to reflect the correct legal references:
- Guidance values for feed: Commission Recommendation 2006/576/EC.
- Maximum levels for food: Commission Regulation (EU) 2023/915.
- IARC Group 3 reworded to: “not classifiable as to its carcinogenicity to humans”.
Thank you for this very important comment regarding legislative aspects.
Comment 3:
All tables: Please indicate that variability expressed as ± is the SD.
Response:
A note has been added below each relevant table to clarify that the variability expressed as ± refers to the standard deviation (SD).
Comment 4:
Table 5: Use crude fiber instead of Cf. The sum of feed components is 87.84%—what is the remaining 12%?
Response:
Thank you for pointing this out. The abbreviation “Cf” has been replaced with “crude fiber”. Regarding the missing percentage (approx. 12%), we have added a clarification in the manuscript stating that the nutritional composition was determined using a near-infrared (NIR) spectroscopy-based analyzer (Foss InfraXact™). This method, while rapid and convenient, does not identify all chemical constituents of the feed. In particular, it may underestimate certain nitrogen-containing compounds and carbohydrate fractions that are not fully detectable in the NIR spectrum. Therefore, the apparent discrepancy is due to unmeasured or undetectable minor feed components.
A corresponding sentence was added in the Materials and Methods section to clarify this limitation of the feed composition data.
Comment 5:
Line 287: Indicate the country for TENEBRIA Sp. z o.o.
Response:
We added the country: “TENEBRIA Sp. z o.o. (Poland)”.
Comment 6:
5.5. Mycotoxin Analysis by LC-MS/MS: Could you provide the detection limit for DON in larvae and frass?
Response:
The limit of detection (LOD) and limit of quantification (LOQ) values for DON were added.
Comment 7:
Since LC-MS/MS equipment was available, it would have been interesting to determine DON metabolites.
Response:
We appreciate this valuable suggestion. The LC-MS/MS system used in our study is a targeted quantification platform, which requires certified reference standards of the analytes to enable accurate detection and quantification. In the present study, we focused on the quantification of parent deoxynivalenol (DON) using available certified standards. The identification and characterization of DON metabolites or degradation products require a non-targeted approach, which we are currently performing using a high-resolution Q-TOF LC-MS system. This ongoing analysis includes screening for metabolic derivatives and biotransformation markers and will be presented in a separate manuscript currently in preparation.
Reviewer 3 Report
Comments and Suggestions for Authors
This manuscript presents an original and interesting study on the bioconversion of deoxynivalenol (DON) mycotoxin in feed by yellow mealworm. The study employs an appropriate LC-MS/MS method to quantify DON concentrations in feed, larvae and their frass and offers a baseline understanding of feed detoxification. The work is timely, methodologically sound, and aligned with broader concerns on feed and food safety. The key contribution is that the presence of DON in the larval diet did not negatively affect biomass gain or survival of the yellow mealworm. In addition, DON did not accumulate in the larvae but was substantially eliminated in their frass.
Key contribution: Perhaps the word ‘demonstrated’ is too categorical and another more appropriate word could be found.
Line 44: Current Commission Recommendation 2006/576/EC set the guidance value (not maximum level) for DON in feed between 0.9 and 12 mg/kg.
Line 45: Current Commission Regulation (EU) 2023/915 set the maximum level for DON in food between 0.15 and 1.75 mg/kg.
Line 46: Group 3 by IARC really means ‘Not classifiable as to its carcinogenicity to humans’.
All tables: Please indicate that variability expressed as ± is the SD.
Table 5: Use crude fiber instead of Cf. The sum of feed components is 87,84 (What could the remaining 12% be?
Line 287: Indicate country for TENEBRIA Sp. z o.o.
5.5. Mycotoxin Analysis by LC-MS/MS: Could you provide the detection limit for DON in larvae and their frass?
Since LC-MS/MS equipment was available, it would have been interesting to determine DON metabolites in the larvae and their excrement.
Author Response
Response to Reviewer 3
Manuscript title:
Bioconversion of Feed Contaminated with Deoxynivalenol Using Yellow Mealworm (Tenebrio molitor) Larvae to Produce Biomass Intended for Feed Purposes
Manuscript ID: toxins-3792158
We would like to thank the Reviewers and the Editor for the time and effort devoted to evaluating our manuscript. We highly appreciate the constructive comments, which helped us to improve the quality, clarity, and scientific merit of the article. Below we provide a detailed point-by-point response to all the comments. All changes made in the revised manuscript are marked in red font.
Comment 1:
The manuscript contains numerous grammatical and spelling errors. It is strongly recommended that a native English speaker or a professional editing service thoroughly review and revise the manuscript.
Response:
As recommended by the Reviewer, the entire manuscript has been thoroughly revised by a professional editing service to eliminate grammatical errors and stylistic inconsistencies, and to improve overall readability and clarity of presentation. The content-related changes are marked in red, and linguistic corrections are marked in blue.
Comment 2:
The wording in the abstract is imprecise and weakens the scientific impact.
Response:
The abstract was revised following a structured scientific format (Background–Objective–Methods–Results–Conclusions).
Comment 3:
Line 86: Please don’t start “The review paper prepared by…”
Response:
We rewrote the sentence for clarity and academic tone.
Comment 4:
I could not see the statistical difference symbol in the tables.
Response:
We added superscript letters (a, b, c) to the tables to denote statistically significant differences and clarified the meaning in table footnotes.
Round 2
Reviewer 1 Report
Comments and Suggestions for Authors
The authors did not respond to my comments; however, I noticed that they made changes and significantly addressed my concerns in the revised manuscript. Moreover, the manuscript still lacks updated and relevant references. It is strongly recommended to include the references mentioned below in the introduction and discussion sections to enhance clarity, relevance, and the overall quality of the manuscript.
- Zhang, Y., Zhang, X., Cao, D., Yang, J., Mao, H., Sun, L.,... Wang, C. (2024). Integrated multi-omics reveals the relationship between growth performance, rumen microbes and metabolic status of Hu sheep with different residual feed intakes. Animal Nutrition, 18, 284-295. doi: https://doi.org/10.1016/j.aninu.2024.04.021
- Wang, Y., Wang, L., Du, Y., Yao, F., Zhao, M., Cai, C.,... Shao, S. (2025). Metabolomics study reveals DON-induced intestinal toxicity in adult zebrafish through disruption of amino acid metabolism and sphingolipid signaling pathway. Aquatic Toxicology, 282, 107324. doi: https://doi.org/10.1016/j.aquatox.2025.107324
- Qiao, L., Zhuang, Z., Wang, Y., Xie, K., Zhang, X., Shen, Y.,... Zhou, S. (2025). Nocturnin promotes NADH and ATP production for juvenile hormone biosynthesis in adult insects. Pest Management Science, 81(6), 3103-3111. doi: https://doi.org/10.1002/ps.8676
- Song, J., Li, W., Gao, L., Yan, Q., Zhang, X., Liu, M.,... Zhou, S. (2024). miR-276 and miR-182013-5p modulate insect metamorphosis and reproduction via dually regulating juvenile hormone acid methyltransferase. Communications Biology, 7(1), 1604. doi: 10.1038/s42003-024-07285-0
Author Response
Dear Reviewer,
We would like to thank you for the time and effort you have devoted to reviewing our manuscript. We appreciate your suggestions aimed at improving the clarity and scientific relevance of our work.
We must admit that we were somewhat surprised by the form of the review, as it did not directly address specific shortcomings of the revised version, while suggesting the inclusion of specific citations. Nevertheless, we carefully assessed all four recommended publications and, after consideration, incorporated two of them — Zhang et al. (2024)and Wang et al. (2025) — as they are the most relevant to the scope of our study:
- Zhang et al. (2024)provides a recent example of applying multi-omics approaches to link diet, microbiota, and metabolic responses, which aligns with our discussion on the possible role of the gut microbiome of molitor in mycotoxin metabolism.
- Wang et al. (2025)offers up-to-date metabolomic insights into the mechanisms of DON toxicity, which strengthens our discussion of potential biochemical pathways involved in DON degradation or excretion.
The remaining two suggested papers (Qiao et al., 2025 and Song et al., 2024) focus on juvenile hormone regulation and microRNA-mediated insect development. While valuable, they do not directly relate to DON toxicity, its metabolism, or feed detoxification in T. molitor, and thus have only limited relevance to our research objectives.
All the changes introduced in response to your comments are marked in green in the revised manuscript for clarity. We believe that these modifications improve the quality of our work while keeping it focused on the main research topic.
Kind regards